# Topiramate Inhibits Capsaicin-Induced Mast Cell Degranulation and CGRP Release in Rat Dura Mater

**DOI:** 10.3390/brainsci14111070

**Published:** 2024-10-27

**Authors:** Raisa Ferreira Costa, Emanuela Paz Rosas, Silvania Tavares Paz, Manuela Figueiroa Lyra de Freitas, Sandra Lopes de Souza, Juliana Ramos de Andrade, Daniella Araújo de Oliveira, Inger Jansen-Olesen, Sarah Louise Christensen, Marcelo Moraes Valença

**Affiliations:** 1Postgraduate Program in Biology Science, Federal University of Pernambuco (UFPE), Recife 50670-901, Brazil; raisa.costa@accamargo.org.br; 2Immunopathology Laboratory (LIKA), Federal University of Pernambuco (UFPE), Recife 50670-901, Brazil; manu_pathy@hotmail.com (E.P.R.); juliana.randrade@ufpe.br (J.R.d.A.); 3Anatomy Department, Federal University of Pernambuco (UFPE), Recife 50670-901, Brazil; silvania_rosas@yahoo.com.br; 4Patology Department, Federal University of Pernambuco (UFPE), Recife 50670-901, Brazil; manuelaflf@uol.com.br; 5Neuropsychiatry and Behavioral Sciences Laboratory, Federal University of Pernambuco (UFPE), Recife 50670-901, Brazil; sanlopesufpe@gmail.com; 6Physiotherapy Department, Federal University of Pernambuco (UFPE), Recife 50670-901, Brazil; dani.aoliveira@ufpe.br; 7Neurology Department, Danish Headache Center, Copenhagen University Hospital—Rigshospitalet, 2600 Copenhagen, Denmark; inger.jansen-olesen@regionh.dk (I.J.-O.); sarah.louise.tangsgaard.christensen@regionh.dk (S.L.C.); 8Anesthesia Department, Critical Care and Pain Medicine, Beth Israel Deaconess Medical Center, Harvard Medical School, Boston, MA 02115, USA; 9Department of Clinical Medicine, University of Copenhagen, 2200 Copenhagen, Denmark; 10Neurosurgery Unit, Federal University of Pernambuco (UFPE), Recife 50670-901, Brazil

**Keywords:** migraine, pathophysiology, CGRP, endogenous substances, mast cells, animal model

## Abstract

Background/Objectives: Migraine is a disease that stands out for its high prevalence and socioeconomic costs. It involves the entire trigeminovascular system, the signaling substances, and their targets. However, the role of meningeal mast cells in migraine is still unclear. To better understand one of the components of neurogenic inflammation underlying migraine pathophysiology, we developed an in vivo rat model in which the dura mater was exposed bilaterally to investigate the influence of topiramate on capsaicin-induced mast cell degranulation and CGRP release from dura mater. Methods: On the day of the experiment, rats were anesthetized, and a craniectomy was performed on each parietal bone. Test substances were applied in situ over the dura mater using the right and left sides of the dura mater for the test and control, respectively. After exposure, the dura mater was processed for mast cell staining and counting. Using this setup, the effect of capsaicin (10^−3^ M) was evaluated in rats of both sexes, and subsequently the effect of in situ (10^−3^ M, 20 µL) and (20 mg/kg/day for 10 days) topiramate treatment on mast cell degranulation and CGRP release were evaluated. Results: In both female and male rats, there was a greater amount of degranulated mast cells in the side stimulated by capsaicin compared to the control side in both females (18 ± 3% vs. 74 ± 3%; *p* = 0.016) and males (28 ± 2% vs. 74 ± 3%, *p* = 0.016). In the group treated with topiramate for 10 days prior to the experiments, capsaicin did not induce mast cell degranulation (control 20 ± 1% vs. capsaicin 22 ± 1%, *p* = 0.375) in contrast to animals treated for 10 days with gavage control (control 25 ± 1% vs. capsaicin 76 ± 1%, *p* = 0.016). Topiramate applied in situ concomitant with capsaicin did not protect the mast cells from degranulation in response to capsaicin (38 ± 2% vs. 44 ± 1%, *p* = 0.016). There was a significant reduction in CGRP release from the dura mater in the group treated with topiramate for 10 days compared to the control. Conclusions: This study demonstrates a novel experimental model wherein systemic administration of topiramate is observed to modulate the impact of capsaicin on meningeal mast cell degranulation.

## 1. Introduction

Two of the most sensitive intracranial structures are the meningeal arteries of the dura mater and large cerebral arteries at the base of the brain [1]. These structures are densely innervated by nociceptive nerve fibers originating from the trigeminal ganglion (TG) containing substance P, calcitonin gene-related peptide (CGRP), pituitary adenylate cyclase-activating peptide (PACAP), and the nitric oxide (NO)-producing enzyme nitric oxide synthase (NOS) [2,3,4,5].

Mast cells are in proximity to meningeal arteries and sensory nerve fibers [6,7]. During degranulation, mast cells release several inflammatory substances [8], causing a sterile neurogenic inflammation of the meninges and activation of trigeminal pain pathways [9,10,11]. Classically, mast cell degranulation has been obtained by administration of the secretagogue agent compound 48/80 either in vivo or in hemi-skull preparations [11,12]. Degranulation of meningeal mast cells has also been shown after infusion of the migraine provoker glyceryl trinitrate (GTN) in a rat in vivo model [12]. The response was significantly inhibited by pretreatment with sumatriptan [12]. Furthermore, the migraine provoker PACAP causes degranulation of rat peritoneal and meningeal mast cells [11,13,14]. Prevention of degranulation via both human and mouse MrgB2/X2 receptors was recently shown to alleviate PACAP-induced tactile hypersensitivity in mice [15]. Collectively, these studies suggest a role of mast cell degranulation in migraine pathophysiology. 

Capsaicin, a TRPV1 channel activator, stimulates sensory nerves to release CGRP and substance P, increasing meningeal blood flow and mast cell degranulation [16,17]. This mechanism captures a broader range of meningeal events and neurogenic inflammation, representing what may occur during spontaneous migraine [10,18,19]. 

Topiramate is effective in the prophylactic treatment of migraine [19]. It has a variety of actions, and its migraine preventive mechanisms of action are not fully elucidated. Among its actions are enhancement of γ-aminobutyric acid (GABA)-mediated inhibition [20,21], inhibition of the AMPA/kainate subtype of glutamate receptors leading to a reduced neuronal excitability [22], a decrease in L-type Ca^2+^ channel currents [23], and inhibition of carbonic anhydrase with the strongest effect on isozymes II and IV [24]. These effects may contribute to topiramate inhibitory effect on CGRP release from the trigeminovascular system [25,26].

In this study, we aimed to develop an in vivo experimental model in which it is possible to investigate the effects on mast cells in an intact biological system by inducing degranulation with capsaicin. We hypothesized that the migraine prophylactic drug topiramate would protect meningeal mast cells from degranulation. Hence, we used the model to study the possible inhibitory effect of in situ and chronic topiramate treatment on capsaicin-induced mast cell degranulation. 

## 2. Materials and Methods

### 2.1. Animals

All experimental procedures were approved by the Ethics Committee on Animal Use (CEUA) of the Federal University of Pernambuco (UFPE) (no: 23076.041888/2018-90). Thirty-five adults (60–70 days of age) Wistar rats (Bioterium José Paulino in the Department of Health Sciences Centers) weighing between 209 g and 298 g were used to complete the study. The rats were kept in an air-conditioned room (23 °C) on a 12/12-h light/dark cycle with food and water ad libitum. They were placed in cages divided into groups with 2–4 rats per cage.

### 2.2. In Vivo Model

On the day of the experiment, the rats were anesthetized with ketamine (1 mg/kg, ip) and xylazine (0.1 mg/kg, ip).

The skin over the head was cut open with a scalpel exposing the skull. With the help of a drill, two cranial windows of 6 mm (rostrocaudal) × 4 mm (mediolateral) were carefully made on the parietal bone to expose the dura mater. The windows were located between the coronal and lambda sutures, slightly below the sagittal line, one on each side of the skull. Following the craniectomies, the preparation was left to rest for five minutes. Synthetic interstitial fluid (SIF) was applied to moisten the exposed dura. Test and control substances were applied in a volume of 20 µL to left and right sides, respectively, for a time of 15 min. SIF was exchanged every 5 min. To avoid leakage, a cotton pad soaked with 60 µL of the solution was used to cover the window during the incubation. 

Immediately after the treatment, the rat was decapitated by a guillotine and the whole head was immersed in a container (collector pot) with 10% paraformaldehyde (PFA) solution. After 24 h of fixation, the dura mater was carefully dissected from the skull, mounted on a microscope slide, and stained with 0.1% toluidine blue for 1 min. After drying, the tissue was protected with coverslips by mounting with Entellan glue. Finally, the number of degranulated and intact mast cells was counted under a light microscope (trinocular with camera) at 400× magnification (Figure 1). 

### 2.3. Experimental Groups

Experiments were performed with rats divided into five groups (n = 7, each group) as follows:(A)In situ capsaicin (females);(B)In situ capsaicin (males);(C)10 days control gavage + in situ capsaicin (males);(D)10 days topiramate gavage + in situ capsaicin (males);(E)In situ topiramate + in situ capsaicin (males).

As no meaningful difference in capsaicin-induced mast cell degranulation was seen between females (group A) and males (group B), subsequent experiments were performed in males to avoid potential interference by the estrous cycle.

#### 2.3.1. In Situ Capsaicin

Topical application of 10^−3^ M, 20 µL capsaicin [27,28,29] (FarmaFórmula, Recife, Brazil) in 0.1% ethanol in SIF was applied to the dura mater via the right side window. The left side dura mater was stimulated with 0.1% ethanol in SIF alone.

#### 2.3.2. Topiramate Treatment

The rats received topiramate (Roval, Recife, Brazil) (20 mg/kg/day) [30,31] by oral gavage in a 1 mL solution of 0.1% ethanol in SIF for 10 days. Control rats received 0.1% ethanol in SIF alone by gavage daily for 10 days.

#### 2.3.3. In Situ Topiramate

For the in situ treatment, 10^−3^ M topiramate and 10^−3^ M capsaicin were applied in a 20 µL solution with a mixture of the two drugs to the right cranial window. The left window was treated with capsaicin alone.

### 2.4. Histological Analyses of Mast Cell Degranulation

The right and left side dura mater samples were stained with acidified 0.1% toluidine blue (pH 2.5) for 1 min. After mounting to microscope slides, two observers blinded to the experimental conditions examined the specimens using a light microscope. Mast cells were considered degranulated when their contents were extensively dispersed by extruded vesicles localized near the cell, or when there was an extensive loss of granule staining outside the cell, and granulated mast cells when the contents were inside the cell. The percentage of degranulated mast cells in the statistical analysis is related to the total number of mast cells observed in the entire dura mater sample.

#### CGRP Measurement

The SIF used to cover the dura mater, preventing it from drying out and allowing fluid to escape from the exposed surface, was collected for a CGRP assay. At the end of the 15-min experiment, the cotton soaked with SIF was pooled together. After vortexing and centrifugation, the supernatant was extracted for analysis using a CGRP ELISA kit (human, Thermo Fisher, Waltham, MA, USA). The rats were anesthetized throughout the experiment, evaluating in vivo CGRP release.

### 2.5. Data Analysis

Mast cell degranulation was quantified as percentage of number of degranulated mast cells out of the total number of mast cells in the test and control sides of dura mater. The data reflect paired samples within individual rats. The data did not meet criteria for parametric testing. Thus, data were analyzed using the Wilcoxon matched pairs signed rank test. Secondary comparisons between independent groups were made by the Mann–Whitney test. GraphPad Prism 9.4.1 was used for analysis. Differences were considered significant when *p* < 0.05. Raw data from all experiments are provided in Table 1.

## 3. Results

### 3.1. Capsaicin-Induced Mas Cell Degranulation in Both Sexes

A small but significant difference was found in basal mast cell degranulation between male and female control sides (Figure 2A). Both female (Figure 2B) and male (Figure 2C) mast cells degranulated in response to capsaicin 74.40 [69.3; 76.1] ± 3% vs. 74.05 [71.4; 78.4] ± 3% vs control side 17.80 [14.9; 24.0] ± 3% and 27.52 [24.5; 30.2] ± 2%, respectively (*p* = 0.016 in both groups). Thus, there was no difference in mast cell degranulation by capsaicin between sexes.

### 3.2. Topiramate Treatment

Oral topiramate treatment alone did not affect mast cell degranulation as the control side of topiramate treated rats had 24.69% [23.0; 26.9] ± 1% vs. 20.37% [16.7; 27.8] ± 1% in water pretreated rats (Figure 3A) (*p* = 0.097). Rats pretreated with water had higher degranulation on the capsaicin side of 75.63% [73.0; 83.2] ± 1% vs. 24.69% [23.0; 26.9] ± 1% on the SIF side (Figure 3B) (*p* = 0.016). Topiramate prevented capsaicin-induced mast cell degranulation as the capsaicin side had a similar degranulation rate 21.93% [19.4; 28.5] ± 1% as the control side had 20.37% [16.7; 27.8] ± 1% degranulation vs on the capsaicin side (Figure 3C) (*p* = 0.375).

### 3.3. In Situ Topiramate Treatment

When the dura mater was treated topically with a mixture of topiramate and capsaicin in an in situ experiment, topiramate did not protect the mast cells from degranulation as compared to treatment with capsaicin alone (Figure 3D), 44.24 [41.8; 48.6] ± 1% vs. 37.50 [26.7; 39.7] ± 2%, respectively, *p* = 0.016. It is noteworthy that the degree of capsaicin-induced mast cell degranulation was lower in this batch of rats than others. The reason for this is unknown.

### 3.4. In Vivo CGRP Release from Dura Mater

There was a significant reduction (*p* = 0.030, Mann–Whitney test) in CGRP release from the dura mater bathed in SIF in the group treated with topiramate for 10 days compared to the control group [35.1 ± 28.5 (n = 6), median 28.9 (95%CI 5.2–65.0), min. 9.4–max. 89.3 vs. 11.2 ± 8.2 pg/mL (n = 5), median 8.5 (95%CI 1.0–21.3), min. 4.9–max. 25.4] (Figure 4).

## 4. Discussion

Using a novel in vivo rat model, this study demonstrates that topiramate, a classic migraine preventive drug, reduces the effect of capsaicin on mast cell degranulation.

This is the first study in which mast cells were studied following dural application of topiramate in vivo. The model has advantages over classical ex vivo/in vitro models since it allows studying the effect of a substance on mast cells in an intact biological system. The use of capsaicin to induce mast cell degranulation is likely to cause more widespread neurogenic inflammation due to the effect on trigeminal nerves in contrast to compound 48/80-induced mast cell degranulation [27]. In the model, we first showed that capsaicin applied directly to dura mater induced mast cell degranulation in both female and male rats. Subsequently, we demonstrated that chronic pretreatment with topiramate prevented mast cell degranulation induced by capsaicin. 

Previous studies have shown CGRP as well as PACAP to be mediators of mast cell degranulation in rat dura mater [13,15,32]. Whereas CGRP does not mediate mast cell degranulation in dura mater from man, PACAP activates the human MrgX2 receptor to mediate mast cell degranulation [15,32]. Histamine and serotonin released from mast cells activate receptors situated at local nerve terminals of sensory neurons [27]. 

Capsaicin increases the release of neuropeptides via TRPV1 activation and causes depletion of sensory neurons after repeated administration as supported by the existing literature [13,33,34]. Only a few other studies, have investigated the effect of capsaicin on mast cell activation. Cheng et al. [35] showed that capsaicin was concentration-dependently related to changes in histamine levels. In the study by Dimtriadou et al. [10], capsaicin (50 mg/kg; was administered subcutaneously in neonatal (48 h of life) rats, which showed no difference in mast cell degranulation between groups. However, in a different treatment design, capsaicin (50 mg/kg, first day of life and 100 mg/kg, 3 days of life), was able to cause the depletion of mast cells thus, the chemical stimulation destroyed the peptidergic nerve endings. This suggests that neonates do not contain a structural and defined neural complex, demonstrating that stress in neonates does not induce mast cell degranulation [36]. Other studies have taken interest in sex differences in mast cell degranulation induced by cholinergic agents (carbachol and nicotine), ovarian hormones, and compound 48/80. Here, the sex of the animal did not influence or interfere in the amount of degranulated mast cells [3,5,37], which is in line with our present findings.

Next, we found that chronic systemic administration of topiramate prevented capsaicin-induced mast cell degranulation, whereas in situ local treatment with topiramate did not prevent capsaicin-induced mast cell degranulation. This suggests that the observations in this study are related to neurogenic inflammation rather than the direct effect of capsaicin and topiramate on mast cells. In migraine patients, the prevention of mast cell degranulation as a derivate reducing neuropeptide release may be one of the mechanisms of the clinical efficacy of topiramate. Topiramate has multiple actions that can act synergistically to contribute to the prevention of migraine. GABA-A-mediated inhibition and changes in ion channels that improve neurotransmission may prevent headaches as it blocks the release of vasoactive peptides and the sensitization of afferent and primary central neurons. Topiramate can modify excitatory neurotransmission through kainate and AMPA receptors, in addition to preventing the synchronous firing of neurons [18,38]. We believe that these mechanisms may contribute to the inhibition of capsaicin-induced mast cell degranulation after chronic systemic administration of topiramate. 

Our results also indicated a significant reduction in CGRP release from the dura mater following chronic topiramate treatment, which may contribute to the decreased mast cell degranulation and, consequently, to the modulation of neurogenic inflammation associated with migraine. This additional action of topiramate on CGRP further supports its potential as a preventive agent in migraine pathophysiology.

In the studies by Storer and Goadsby, an electrode was used to evoke dilation of blood vessels bilaterally in the skull and topiramate inhibited subsequent cell firing in the trigeminocervical complex (TCC) at a dose of 30 mg kg^−1^ in cats [39]. In another study, topiramate dose-dependently (3, 5, 10, 30, and 50 mg/kg) inhibited activation of primary trigeminal afferents subsequent to electrical stimulation of superior sagittal sinus [39]. Topiramate at a dose of 30 mg/kg also significantly inhibited capsaicin-induced CGRP expression in adult rats [40]. These studies corroborate with our results, as a dose of 20 mg/kg of topiramate prevented mast cell degranulation induced by capsaicin. The close connection of mast cells with sensory neurons was suggested as an important element of the neurogenic inflammation model of migraine [41,42,43].

CGRP released during the activation of sensory nerves in the trigeminovascular system plays a crucial role in neurogenic inflammation and mast cell degranulation, promoting vasodilation. While the literature shows that inhibiting CGRP or its receptors reduces the frequency and severity of migraine attacks, our findings suggest that topiramate may exert a similar effect by reducing mast cell degranulation and CGRP release after chronic exposure [44,45,46]. This modulation by topiramate reinforces its preventive action in migraine, aligning with therapies that target CGRP directly, such as CGRP receptor antagonists and monoclonal antibodies.

Our study aligns with this perspective, suggesting that capsaicin induces mast cell degranulation through the release of sensory neuropeptides, without directly activating receptors on mast cells. 

### Strengths and Limitations

A major strength of the study is that the animal serves as its own control to avoid bias by variation in basal mast cell degranulation caused by either intrinsic mechanisms, surgical procedures, or handling of tissue. The low degree of mast cell degranulation on the control side leaves a wide window for measuring the degree of mast cell degranulation after an intervention. Importantly, the mast cell counting was blinded and performed independently by two different observers (RC and EPR). Both female and male rats had a strong degranulation response to capsaicin and, therefore, subsequent experiments were performed on male rats. This is a limitation as we cannot know if there would be a sex difference in the response to topiramate. In the in situ topiramate experiment, the degree of capsaicin-induced mast cell degranulation was much lower than observed in the other experiments. We do not have an explanation for this, but such batch-to-batch variation stresses the importance of having the rats as its own control, or alternatively always run the test and control experiments simultaneously. Another limitation is related to in situ inflammation after craniectomy and the chance for mechanical induction of mast cell degranulation during the craniectomy. The low level of degranulation on the control side indicates that this was not a major problem and again variation should be limited by having the animal as its own control. Additional limitation is the absence of immunostaining for TRPV1 on dural mast cells and the interpretation of capsaicin’s effects.

Hormonal fluctuations in females may significantly influence pain sensitivity and mast cell activation, potentially leading to differential responses to treatments such as topiramate. However, future studies should include both sexes to better understand the mechanisms involved and how they may affect treatment efficacy. In addition, the observed batch-to-batch variability in mast cell degranulation may affect the reproducibility of the results, highlighting the need to carefully consider these factors in experimental design. Addressing these issues will increase the robustness and clinical relevance of future research.

In this study, capsaicin was administered at a concentration of 10^−3^ M. This dose is justified by evidence showing that 10^−3^ M capsaicin effectively activates TRPV1 channels, a key mechanism in pain modulation. Topiramate was evaluated under two treatment conditions: an in situ dose of 10^−3^ M, based on its immediate pharmacological effects, and a treatment of 20 mg/kg/day for 10 days, which reflects its longer-term therapeutic potential in altering neuronal excitability and preventing pain over sustained periods. However, these mechanisms remain speculative, as they could not be confirmed in the current experiment. Further complementary studies are required to validate these findings and fully understand the effects of capsaicin and topiramate in this context.

## 5. Conclusions

This novel in vivo model to study dural mast cell degranulation proved to be reproducible and with low numbers of degranulated mast cells in control specimens. We found that the TRPV1 agonist capsaicin induced mast cell degranulation in vivo and that chronic pretreatment with topiramate prevented degranulation of dural mast cells following topical exposure to capsaicin.

In this study, it became evident that topiramate can reduce mast cell degranulation in the dura mater and decrease CGRP release in situ, suggesting that the preventive actions of topiramate in migraine may occur through these mechanisms, at least in part. This mechanism is also observed with other agents used in migraine prevention, such as anti-CGRP antibodies and CGRP antagonists.

CGRP plays a crucial role in the pathophysiology of migraine. It is a neuropeptide widely distributed in the nervous system, particularly in sensory neurons. During a migraine attack, levels of CGRP increase, leading to the dilation of blood vessels and the transmission of pain signals. CGRP also contributes to neurogenic inflammation by promoting the release of inflammatory mediators and causing mast cell degranulation.

The release of CGRP in the trigeminovascular system is associated with the activation of trigeminal pain pathways, a key component in the onset and progression of migraine. Because of its significant role, CGRP has become a major target for migraine therapies, with CGRP receptor antagonists and monoclonal antibodies designed to block CGRP or its receptors, effectively reducing the frequency and severity of migraine attacks.

## Figures and Tables

**Figure 1 brainsci-14-01070-f001:**
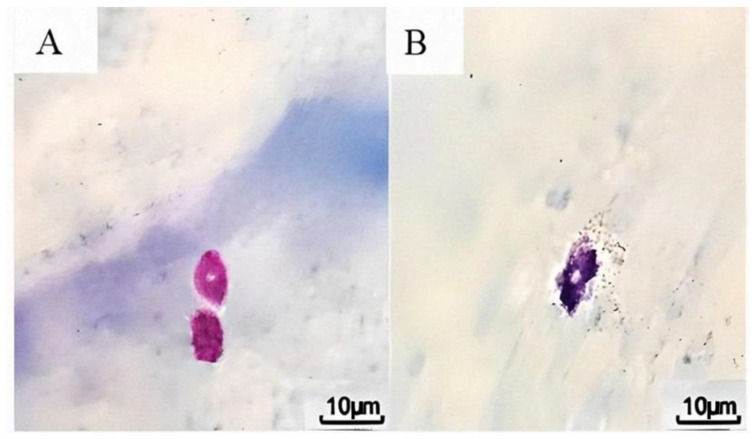
Microscopic visualization of mast cells in toluidine blue-stained dura mater. (**A**) Intact mast cells in dura mater. (**B**) Degranulated mast cell in dura mater. Scale bar = 10 µm.

**Figure 2 brainsci-14-01070-f002:**
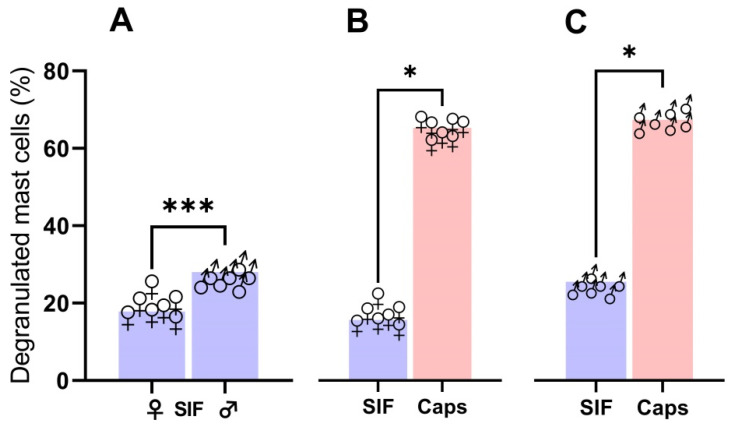
The degranulating effect of capsaicin on mast cells in dura mater after treatment with vehicle (left side window) and capsaicin (right side window) in female (n = 7) and male (n = 7) rats. (**A**) Males had a significantly higher degree of degranulation on the control side than females, Mann–Whitney, *p* < 0.001 ***. In both females (**B**) and males (**C**), capsaicin induced mast cell degranulation, Wilcoxon matched pairs signed rank test *p* = 0.016 * females = ♀; males = ♂.

**Figure 3 brainsci-14-01070-f003:**
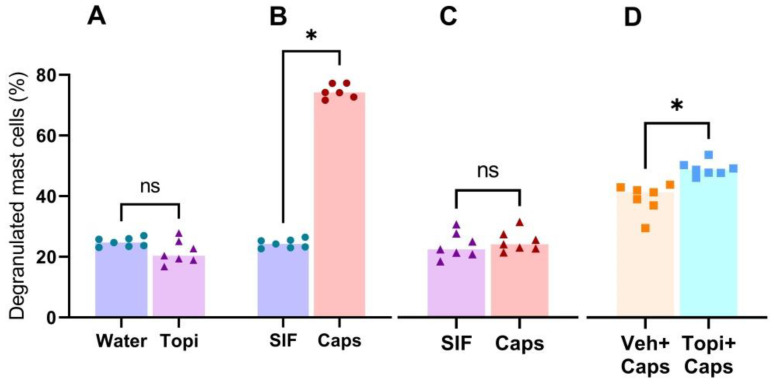
The effect of capsaicin in male rats treated orally with topiramate for 10 days as compared to water control. (**A**) Topiramate treatment did not influence the basal level of mast cell degranulation in control side dura mater samples, Mann–Whitney test, *p* = 0.097. (**B**) Rats treated with water for 10 days prior to experiments had normal mast cell degranulation response to capsaicin, Wilcoxon matched pairs signed rank test, *p* = 0.016. (**C**) Chronic topiramate treatment prior to capsaicin exposure protected the mast cells from degranulation, Wilcoxon matched pairs signed rank test, *p* = 0.375. (**D**) In situ application of topiramate did not protect mast cells from capsaicin-induced degranulation, Wilcoxon matched pairs signed rank test, *p* = 0.016 * Gavage water = circles, gavage topiramate = triangles, and in situ topiramate = squares, ns = no significative.

**Figure 4 brainsci-14-01070-f004:**
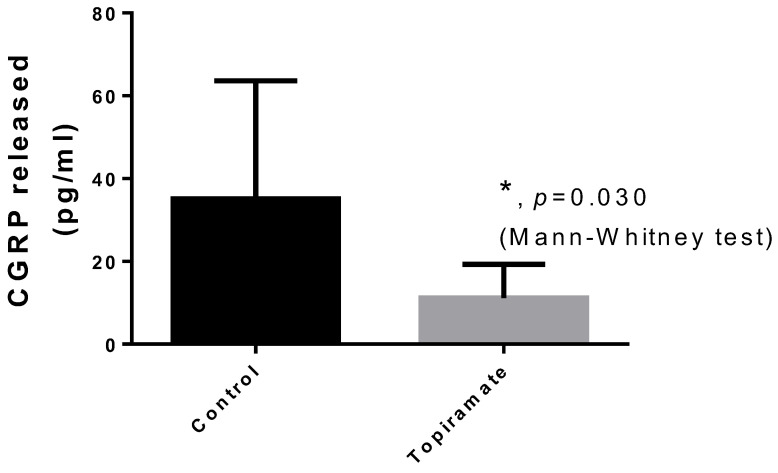
Significant reduction in CGRP release from the dura mater bathed in SIF was observed in the group treated with topiramate for 10 days compared to the control group.

**Table 1 brainsci-14-01070-t001:** Raw data for all animals into the five groups: raw numbers degranulated and non-degranulated mast cells.

	SIF			Capsaicin		
Females In Situ Capsaicin	Total Cell Count	Degranulated Cell Count	%	Total Cell Count	Degranulated Cell Count	%
Rat 1	191	34	17.80	212	147	69.30
Rat 2	204	34	16.70	209	147	70.40
Rat 3	102	20	19.60	161	120	74.60
Rat 4	235	35	14.90	148	110	74.40
Rat 5	182	29	15.93	168	120	71.50
Rat 6	142	34	23.94	161	124	76.10
Rat 7	143	28	19.58	159	120	75.50
Males in situ capsaicin						
Rat 1	278	80	26.11	260	200	76.92
Rat 2	234	60	25.64	212	157	74.05
Rat 3	215	65	30.23	176	138	78.40
Rat 4	200	49	24.50	224	160	71.42
Rat 5	195	55	28.21	237	180	75.94
Rat 6	139	40	28.78	166	120	72.28
Rat 7	218	60	27.52	191	140	73.29
10 days Control Gavage						
Rat 1	219	59	26.94	193	141	73.05
Rat 2	162	40	24.69	162	120	74.07
Rat 3	202	57	25.75	231	182	78.78
Rat 4	181	47	25.96	229	173	75.54
Rat 5	260	60	23.07	209	174	83.25
Rat 6	230	50	23.40	238	180	75.63
Rat 7	253	60	23.71	249	196	78.71
	Capsaicin			Capsaicin + Topiramate		
In situ topiramate						
Rat 1	239	64	26.77	419	204	48.68
Rat 2	480	180	37.50	368	168	45.65
Rat 3	350	125	35.41	538	240	44.60
Rat 4	420	160	38.08	323	140	43.34
Rat 5	256	86	33.59	231	100	43.29
Rat 6	305	115	38.98	226	100	44.24
Rat 7	269	107	39.78	172	72	41.86

## Data Availability

Raw data is provided in the manuscript. Photographs of samples are available upon reasonable request to the corresponding author.

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
