# Peer review of "Topiramate Inhibits Capsaicin-Induced Mast Cell Degranulation and CGRP Release in Rat Dura Mater"

_brainsci, 2024, doi:10.3390/brainsci14111070_

Round 1
Reviewer 1 Report
Comments and Suggestions for Authors
This study provides valuable insights into the role of topiramate in modulating neurogenic inflammation in migraine models.
With minor revisions to improve clarity and methodological justification, the manuscript would significantly contribute to the field. I recommend acceptance with minor revisions to address these issues.
Suggestions:
Refine the background section to eliminate unclear phrases and provide a more compelling rationale.
Justify the concentrations and dosages used for capsaicin and topiramate.
Emphasize the novelty and potential clinical implications of the findings in the conclusion.
Consider rephrasing parts of the methods and results for enhanced clarity and flow.
By addressing these points, the manuscript will present a stronger and more comprehensive narrative of the research conducted.
Author Response
Dear Editor and reviewers,
We sincerely appreciate your valuable feedback and suggestions. We have carefully considered and addressed each comment. Below, we provide detailed responses to each point (responses indicated in red) and references to the corresponding sections of the revised manuscript where the relevant changes have been made.
Responses to review 1
- Refine the background section to eliminate unclear phrases and provide a more compelling rationale.
Reworded lines 62-65 of paragraph 3 on page 2 to make the phrasing clear.
“Capsaicin, a TRPV1 channel activator, stimulates sensory nerves to release CGRP and substance P, increasing meningeal blood flow and mast cell degranulation [16,17] . This mechanism captures a broader range of meningeal events and neurogenic inflammation, representing what may occur during spontaneous migraine [10,18,19] .”
2. Justify the concentrations and dosages used for capsaicin and topiramate.
We have included citations to the reference articles for the dosages (page 3).
3. Emphasize the novelty and potential clinical implications of the findings in the conclusion.
We have emphasized on page 9, paragraphs 2 and 3, lines 296-312.
“Hormonal fluctuations in females may significantly influence pain sensitivity and mast cell activation, potentially leading to differential responses to treatments such as topiramate. However, future studies should include both sexes to better understand the mechanisms involved and how they may affect treatment efficacy. In addition, the observed batch-to-batch variability in mast cell degranulation may affect the reproducibility of the results, highlighting the need to carefully consider these factors in experimental design. Addressing these issues will increase the robustness and clinical relevance of future research.
In this study, capsaicin was administered at a concentration of 10-3 M. This dose is justified by evidence showing that 10-3 M capsaicin effectively activates TRPV1 channels, a key mechanism in pain modulation. Topiramate was evaluated under two treatment conditions: an acute dose of 10-3 M, based on its immediate pharmacological effects, and a chronic regimen of 20 mg/kg/day for 10 days, which reflects its longer-term therapeutic potential in altering neuronal excitability and preventing pain over sustained periods. However, these mechanisms remain speculative, as they could not be confirmed in the current experiment. Further complementary studies are required to validate these findings and fully understand the effects of capsaicin and topiramate in this context.”
4. Consider rephrasing parts of the methods and results for enhanced clarity and flow.
We have rewritten portions of the methods and results for greater detail (see page 4 until page 7, from paragraph 2.3.1, lines 124 to paragraph 3.3, line 196).
We hope that our revisions address all of your concerns and improve the quality of the manuscript. We appreciate your time and consideration, and we look forward to your feedback.
Best regards,
Raisa Costa

Reviewer 2 Report
Comments and Suggestions for Authors
The study introduces a unique in vivo rat model to study topiramate's effects on capsaicin-induced mast cell degranulation, filling an important gap in migraine research. The dual approach of studying both acute and chronic effects of topiramate is well-justified, adding depth to the understanding of its mechanism of action. The methodology is clearly described, with a well-justified use of controls and experimental groups. Overall, the study potentially offers new insights into how topiramate could prevent neurogenic inflammation, a key factor in migraine pathophysiology.
The authors are encouraged to consider the following points in revising the manuscript:
While the CGRP release results are significant, the discussion does not go deeply into the broader implications of reduced CGRP beyond its connection to mast cell degranulation.
The decision to focus on male rats after initial experiments with both sexes is acknowledged, but the absence of further sex-based analysis could be a limitation, especially in migraine research where sex differences are well-documented.
While the discussion summarizes key findings, it lacks broader comparisons to related studies on topiramate and capsaicin. The potential clinical implications of the findings, particularly in terms of migraine prevention, could be elaborated more.
A more thorough discussion of CGRP's role in migraine pathophysiology and how the study's findings may align with or differ from existing knowledge would strengthen the conclusions.
The study could benefit from discussing the possible clinical translation of these findings in more detail, particularly how the findings might influence future migraine treatments.
Given the unexpected batch-to-batch variation in mast cell degranulation, considering alternative or complementary assays to confirm findings might improve robustness.
The limitations section is well-structured but could be more detailed regarding the potential implications of batch variations and the choice to focus on male rats.
In conclusion, this paper is strong in its design and methodology, but expanding the discussion and addressing some key limitations could enhance its clarity and quality.
Author Response
Dear Editor and reviewers,
We sincerely appreciate your valuable feedback and suggestions. We have carefully considered and addressed each comment. Below, we provide detailed responses to each point (responses indicated in red) and references to the corresponding sections of the revised manuscript where the relevant changes have been made.
Responses to review 2
- The study introduces a unique in vivo rat model to study topiramate's effects on capsaicin-induced mast cell degranulation, filling an important gap in migraine research. The dual approach of studying both acute and chronic effects of topiramate is well-justified, adding depth to the understanding of its mechanism of action. The methodology is clearly described, with a well-justified use of controls and experimental groups. Overall, the study potentially offers new insights into how topiramate could prevent neurogenic inflammation, a key factor in migraine pathophysiology.
The authors are encouraged to consider the following points in revising the manuscript:
While the CGRP release results are significant, the discussion does not go deeply into the broader implications of reduced CGRP beyond its connection to mast cell degranulation.
The decision to focus on male rats after initial experiments with both sexes is acknowledged, but the absence of further sex-based analysis could be a limitation, especially in migraine research where sex differences are well-documented.
While the discussion summarizes key findings, it lacks broader comparisons to related studies on topiramate and capsaicin. The potential clinical implications of the findings, particularly in terms of migraine prevention, could be elaborated more.
A more thorough discussion of CGRP's role in migraine pathophysiology and how the study's findings may align with or differ from existing knowledge would strengthen the conclusions.
The study could benefit from discussing the possible clinical translation of these findings in more detail, particularly how the findings might influence future migraine treatments.
Given the unexpected batch-to-batch variation in mast cell degranulation, considering alternative or complementary assays to confirm findings might improve robustness.
The limitations section is well-structured but could be more detailed regarding the potential implications of batch variations and the choice to focus on male rats.
In conclusion, this paper is strong in its design and methodology, but expanding the discussion and addressing some key limitations could enhance its clarity and quality.
We have expanded the discussion on the role of CGRP in the pathophysiology, highlighting the implications of its reduction and its connection with mast cell degranulation. “CGRP released during the activation of sensory nerves in the trigeminovascular system plays a crucial role in neurogenic inflammation and mast cell degranulation, promoting vasodilation. While the literature shows that inhibiting CGRP or its receptors reduces the frequency and severity of migraine attacks, our findings suggest that topiramate may exert a similar effect by reducing mast cell degranulation and CGRP release after chronic exposure [43–45] . This modulation by topiramate reinforces its preventive action in migraine, aligning with therapies that target CGRP directly, such as CGRP receptor antagonists and monoclonal antibodies. Additionally, we have incorporated broader comparisons with related studies on topiramate and capsaicin to strengthen the context of our findings (that the adjustments made to the discussion improve this issue, please re-read the entire discussion starting on page 7). Finally, we further elaborated on the potential clinical implications of these results (see page 9, paragraphs 2 and 3).
We hope that our revisions address all of your concerns and improve the quality of the manuscript. We appreciate your time and consideration, and we look forward to your feedback.
Best regards,
Raisa Costa
